# Role of Neutrophil Extracellular Traps in COVID-19 Progression: An Insight for Effective Treatment

**DOI:** 10.3390/biomedicines10010031

**Published:** 2021-12-23

**Authors:** María Amparo Blanch-Ruiz, Raquel Ortega-Luna, Guillermo Gómez-García, Maria Ángeles Martínez-Cuesta, Ángeles Álvarez

**Affiliations:** 1Departamento de Farmacología, Facultad de Medicina y Odontología, Universidad de Valencia, 46010 Valencia, Spain; Maria.a.blanch@uv.es (M.A.B.-R.); orlura@uv.es (R.O.-L.); guigogar@alumni.uv.es (G.G.-G.); 2Centro de Investigación Biomédica en Red Enfermedades Hepáticas y Digestivas (CIBERehd), 46010 Valencia, Spain

**Keywords:** COVID-19, SARS-CoV-2, neutrophils, neutrophil extracellular traps, NETs, cytokine storm, acute respiratory distress syndrome, ARDS

## Abstract

The coronavirus disease 2019 (COVID-19), caused by SARS-CoV-2, has resulted in a pandemic with over 270 million confirmed cases and 5.3 million deaths worldwide. In some cases, the infection leads to acute respiratory distress syndrome (ARDS), which is triggered by a cytokine storm and multiple organ failure. Clinical hematological, biochemical, coagulation, and inflammatory markers, such as interleukins, are associated with COVID-19 disease progression. In this regard, neutrophilia, neutrophil-to-lymphocyte ratio (NLR), and neutrophil-to-albumin ratio (NAR), have emerged as promising biomarkers of disease severity and progression. In the pathophysiology of ARDS, the inflammatory environment induces neutrophil influx and activation in the lungs, promoting the release of cytokines, proteases, reactive oxygen species (ROS), and, eventually, neutrophil extracellular traps (NETs). NETs components, such as DNA, histones, myeloperoxidase, and elastase, may exert cytotoxic activity and alveolar damage. Thus, NETs have also been described as potential biomarkers of COVID-19 prognosis. Several studies have demonstrated that NETs are induced in COVID-19 patients, and that the highest levels of NETs are found in critical ones, therefore highlighting a correlation between NETs and severity of the disease. Knowledge of NETs signaling pathways, and the targeting of points of NETs release, could help to develop an effective treatment for COVID-19, and specifically for severe cases, which would help to manage the pandemic.

## 1. Introduction

The infection caused by severe acute respiratory syndrome coronavirus 2 (SARS-CoV-2) was designated by the World Health Organization as coronavirus disease 2019 (COVID-19) when it emerged in 2019 in Wuhan (China) [1]. The virus SARS-CoV-2 belongs to a family of single positive-strand RNA [2] viruses that infect vertebrates by entering the host cell through the S protein (‘spike’), which is cleaved into two subunits—S1 and S2—by a host cellular furin protease [3,4]. Reports suggest that the S1 subunit is responsible for binding to the human angiotensin-converting enzyme II (ACE2), a metallopeptidase that mediates the process of virus entry [5,6] into the host cells by endocytosis, while the S2 subunit is responsible for the fusion of the membranes of the virus with those of the host cells [3]. The participation of both subunits in the virus’s entry into the host cells initiates the cycle of viral replication.

COVID-19 has represented an unprecedented challenge to many health systems around the world due to the efficient person-to-person transmission [7], the frequently (15–30% of cases) severe complications that require hospital admission, and the lack of a specific treatment to resolve them [8]. In some cases, SARS-CoV-2-infected patients—who develop severe respiratory distress syndrome (ARDS) [9,10] and pneumonia—eventually die. In addition, SARS-CoV-2 can infect other organs such as the heart, liver, brain, and kidneys [11], and generate a systemic immune overreaction response [12]. Moreover, the severity of COVID-19 disease is associated with age, biological sex, and comorbidity factors such as diabetes, obesity, and chronic lung and cardiovascular diseases [9,13,14,15].

The lack of selective and efficient antiviral drugs against SARS-CoV-2 has made vaccines fundamental in the prevention of COVID-19. Nevertheless, the incomplete effectiveness of the available vaccines due to numerous more infectious SARS-CoV-2 mutations, and the failure to achieve universal vaccination, explain the continuous increases in the number of COVID-19 patients, with over 270 million confirmed cases worldwide and over 5.3 million deaths [16]. Therefore, there is an urgent challenge for clinicians who are struggling to manage the severe symptoms of patients with the worst prognosis, currently identified as “the immune overreaction kills” [12]. The present study reviews the latest clinical studies of the initial biomarkers associated with fatal COVID-19 progression. Specifically, it focuses on the source of immune cells and the signaling pathways of the inflammatory and biochemical mediators involved in multiorgan failure, with the intention of identifying new therapeutic targets to prevent the progression of the illness. In order to find an answer to these questions, we carried out searches (until August 2021) of the databases PubMed and ClinicalTrials, both from the National Library of Medicine (in the English language only). In particular, to evaluate the importance of biomarkers, neutrophils, and neutrophil extracellular traps (NETs) in COVID-19, the following search terms were used in PubMed: 1. (COVID-19[Title/Abstract]) AND (cytokine storm[Title/Abstract]) AND (biomarkers[Title/Abstract]); 2. ((COVID-19[Title/Abstract]) OR (SARS-CoV-2[Title/Abstract])) AND ((neutrophil[Title/Abstract]) OR (neutrophil extracellular traps[Title/Abstract])); 3. ((COVID-19[Title/Abstract]) OR SARS-CoV2[Title/Abstract])) AND (thrombosis[Title/Abstract]) AND (neutrophil[Title/Abstract])). 

In order to identify (in ClinicalTrials) the drugs related to NETs that are currently the subject of a clinical trial, we employed the following key words in the search field “condition or disease”: COVID, SARS-CoV-2, Coronavirus disease 2019, Severe Acute Respiratory Syndrome Coronavirus 2, Novel Coronavirus, 2019-nCoV, Coronavirus Disease 19, SARS Coronavirus 2, and Wuhan Coronavirus. We then selected “Interventional (Clinical Trial)” as the type of study, obtaining 3520 results. Next, we filtered the results by introducing the following drug names in the “interventions” column: colchicine, disulfiram, metformin, alvelestat, fostamatinib, Vitamin D, Vitamin D3, 25-Hydroxivitamin D3, cholecalciferol, N-acetylcysteine, dornase alfa, pulmozyme. These search terms rendered a total of 92 results, on which we have based the present study. 

## 2. Biomarkers in COVID-19

The number of patients with COVID-19 continues to rise, and, although most cases are mild or asymptomatic, the paradigm of the disease’s progression highlights the role of inflammation and thrombotic events as the main cause of death.

The acute progression of the disease in symptomatic patients starts with an early infection phase with mild symptoms, during which the virus infiltrates the lung parenchyma and triggers the innate immune response mediated by monocytes and macrophages. In addition, early infection is characterized by neutrophilia, exhibited as an increase in the neutrophil-to-lymphocyte ratio (NLR) and neutrophil-to-albumin ratio (NAR) (Figure 1) [17,18,19,20]. Around day five of infection, the collateral tissue injury and the inflammatory process that follows (vasodilation, endothelial permeability, leukocyte recruitment, and neutrophil infiltration) lead to further pulmonary damage, hypoxemia, and cardiovascular stress, with patients presenting abnormal chest imaging. During diagnosis, it is typical to observe the following patterns: increased whiteness or ground-glass opacities of the lungs, consolidation, a reticular pattern, and a crazy paving pattern, while other manifestations in chest imaging, such as airway and pleural alterations, fibrosis, and the presence of nodules are atypical, but can also be observed in patients [21] (Figure 1). Alarmingly, in some patients, the host inflammatory response continues to amplify, resulting in an exaggerated systemic inflammation, or cytokine storm, which is the hallmark of the severity of the disease. This phase is characterized by multiple organ failure and elevation of key inflammatory markers (Figure 1) [22]. In this context, clinical data have shown a diversity of biomarkers associated with the progression of COVID-19 disease, including hematologic, biochemical, coagulation and inflammatory markers [22]. Thus, there is a growing body of evidence of the presence of biomarkers of organ damage in the blood of these severely affected patients, specifically in the liver (alanine and aspartate aminotransferase (ALT and AST)), kidney (bilirubin, blood urea nitrogen, and creatinine), and heart (creatine kinase (CK-MB), myoglobin, and cardiac troponin I), and also of multifunctional organ damage indicators (lactate dehydrogenase (LDH)) [22]. These changes are accompanied by an increase in coagulation markers, such as prothrombin time and D-dimer, and inflammatory markers, such as erythrocyte sedimentation rate (ESR), C-reactive protein (CRP), serum ferritin, procalcitonin (PCT), interleukins (IL-1β, IL-2, IL-6, IL-7, IL-8, IL-10, IL-17), interferon γ (IFNγ), monocyte chemoattractant protein 1 (MCP-1), granulocyte colony-stimulating factor (G-CSF), and tumor necrosis factor α (TNF-α) (Figure 1) [22,23,24,25,26]. In addition, a strong correlation has been described between the abovementioned parameters and a rise in white blood cells, together with a decrease in the number of lymphocytes, platelets, eosinophils, T-cells, B-cells, and natural killer cells [22].

The origin of this cytokine storm related with the severity of COVID-19 disease needs to be investigated to identify therapeutic tools to fight against the pandemic. Current research is focused on circulating active neutrophils as the source, not only of the release of cytokines, but also of the formation of NETs involved in lung tissue damage (Figure 1), which have been detected in SARS-CoV-2-infected patients. NETs are networks of extracellular fibers that are primarily composed of DNA and that were first described to bind to pathogens and kill them [27]. Excessive NETs formation has been shown to induce the immune-thrombotic state observed in sepsis, ARDS, and cancer [28]. 

## 3. Neutrophils and Neutrophil Extracellular Traps in COVID-19 

As we have seen in the previous section, COVID-19 is associated with a cytokine storm, triggering an increase in plasma concentrations of diverse interleukins, such as IL-1β, IFNγ, MCP-1, and TNF-α [23,24,25,26]. These inflammatory mediators can regulate neutrophil function and their infiltration to the inflammatory focus. This cytokine storm promotes a signaling loop between macrophages and neutrophils that may lead to the prolonged inflammatory status seen in severe COVID-19 patients [12]. In fact, a transcriptomic profile study based on the GSE1739 dataset has demonstrated, by means of gene ontology (GO) analysis, that neutrophil activation and degranulation are the most activated processes in SARS infection, and are identified with the GO terms GO0002283 and GO0043312, respectively [29]. Likewise, neutrophilia has been described as an indicator of severe respiratory symptoms and poor outcome in patients with COVID-19 [17,18,19,30]. In this sense, neutrophil count, NLR, and NAR appear to be potential clinical markers of COVID-19 progression. Although a cut-off value of >3.4 × 10^9^ neutrophils/L initially distinguished between severe and non-severe COVID-19 patients—>3.42 × 10^9^ neutrophils/L was determined in one study (61 patients, 42 mild cases and 19 severe cases; area under receiver operation characteristic (AUROC): 0.728, 95% confidence interval (95% CI): 0.605–0.851, *p* = 0.0046, with 100% of sensitivity and 45.2% of specificity) [18], while that of 3.65 × 10^9^ neutrophils/L was estimated in a meta-analysis of 27 studies, including a total of 4117 COVID-19 patients (2950 non-severe cases and 1167 severe cases; AUROC: 0.79, *p* = 0.004, with 75% of sensitivity and 81.2% of specificity) [19]—later studies observed that other clinical markers, such as a lower lymphocyte count, may be more useful for the early identification of severe COVID-19 patients [31,32,33,34,35,36]. These studies, including some meta-analyses, have analyzed the parameter of lymphocyte count as a predictor of COVID-19 severity. For example, the study by Tahtasakal et al., which included 534 hospitalized patients (398 of whom were mild–moderate at admission) estimated an optimal cut-off value of <1.04 × 10^9^ lymphocytes/L (sensitivity: 63.24% and specificity: 67.34%, AUROC: 0.678, 95% CI: 0.636–0.717) [33], while another meta-analysis by Elshazli et al. demonstrated that patients with normal lymphocyte count were less likely to develop severe illness, to be admitted to an intensive care unit (ICU), or to die. In said meta-analysis, disease severity was based on sixteen studies with 680 mild patients and 1128 severe patients, with an estimated odds ratio (OD): 0.30, 95% CI: 0.19–0.47, and *p* < 0.001; ICU admission was based on four studies with 73 floor patients and 207 ICU patients, with an estimated OD: 0.23, 95% CI: 0.09–0.62 and *p* < 0.003; mortality was based on 7 studies with 756 live patients and 424 deceased patients, with an estimated OD: 0.21, 95% CI: 0.10–0.47, and *p* < 0.001) [36]. In this context, several studies have reported that a high NLR value predicts the severity of the disease in the early stage of SARS-CoV-2 infection [18,35,37,38]. In line with this, increased NLR has been considered an independent risk factor for mortality in hospitalized patients and a predictor of the severity of the illness and of a longer stay in hospital. It has even been associated with worse immune function and prolonged virus clearance, especially in patients with other comorbidities, such as diabetes [38,39,40,41,42,43,44]. Hence, a cut-off value of NLR > 3.13 may help clinicians to identify patients that are likely to require more intensive monitoring and clinical care [44]. Moreover, another clinical marker involving neutrophils, NAR, has been described as a new predictor of mortality in COVID-19 patients, since low albumin levels in acute infections are associated with mortality risk in hospitalized patients [38]. A ROC analysis of a total of 144 patients—divided into non-critical and critical groups—has recently related a NAR value of 201.5 with mortality in all patients with COVID-19 with a sensitivity of 71.7% and a specificity of 71.7% (AUCROC: 0.736, 95% CI: 0.641–0.832, *p* < 0.001), though it must be pointed out that age differed significantly between the two groups (non-critical patients: 62.0 ± 14.3 years, critical patients: 68.6 ± 12.2 years, *p* < 0.004) [20]. Altogether, neutrophilia, NLR, and NAR in the early stages of the infection are correlated with COVID-19 severity (Figure 1).

Furthermore, despite these ratios, an increase in neutrophils has not only been reported in blood, but also in lung tissue [45,46,47]. Neutrophil infiltration in pulmonary capillaries with extravasation to the alveolar space has been observed in lung autopsies obtained from patients who died from COVID-19, indicating inflammation in the entire lower respiratory tract. Although neutrophils can play a protective role in response to the infection, extensive and uncontrollable activation of these leukocytes can have detrimental effects and result in pneumonia and/or ARDS [48,49,50]. In fact, these two disorders are the main complications in COVID-19 patients [9]. ARDS is an acute syndrome that can be the result of pneumonia or another lung insult, and it is characterized by non-cardiogenic respiratory failure and bilateral opacity in the lungs, which leads to oxygenation impairment [51] (Figure 1). The treatment for this condition is still only supportive, based mainly on lung-protective mechanical ventilation, neuromuscular blockade, prone position, and conservative fluid administration [51]. ARDS has a mortality rate of 30–50%, which highlights the need for new and effective therapeutic approaches [52,53]. This pathology is associated with diffuse lung inflammation, endothelial and epithelial damage, and an increased vascular permeability, which triggers the disruption of the integrity of the endothelial barrier and subsequently leads to diffuse lung damage [54,55]. Many cell types, including endothelial and alveolar epithelial cells, alveolar macrophages, monocytes, neutrophils, and platelets, are involved in the pathophysiology of ARDS [56]. In the early stages of infection, neutrophils and other phagocytizing cells are recruited to the microvasculature, where they interact with platelets and release a mixture of defensins, chemokines, and cytokines. In such an inflammatory environment, high concentrations of inflammatory mediators present in the alveolar space lead to the influx of neutrophils with their subsequent activation, therefore promoting the release of reactive oxygen species (ROS), proteases, cytokines, and NETs, and potentiating alveolar injury (Figure 1) [55,56]. In the respiratory tract, NETosis bolsters the protection against infection by increasing the viscosity of the mucus and by destroying the pathogens [57]. Nonetheless, at the same time, NETosis promotes the development of several infectious complications of the lungs, including ARDS [57]. In this regard, a recent clinical trial with 310 patients (median age of 75 years, 64% male, 23.2% with a smoking history, 17.4% with chronic obstructive pulmonary disease, 21.9% pre-treated with an antibiotic, 78.5% with at least two systemic inflammatory response syndrome criteria, and 53.6% with severe pneumonia) highlighted the correlation between levels of NETs and the severity of pneumonia. Pneumonia severity was measured as: 1) the time required for the clinical stabilization of vital signs at two consecutive measurements ≥ 12 h apart (TTCS), 2) the time to effective discharge from hospital, and 3) mortality. Patients in the highest AUC NETs quartile (fourth quartile) had longer TTCS (5.0 days, interquartile range (IQR): 2.6–9.0 vs 4.0 days, IQR: 2.0–7.4 from first to third quartiles, *p* = 0.041) and longer median time to effective discharge from hospital (9.0 days, IQR: 5.0–14.0 vs. 7.0 days, IQR: 5.0–11 from first to third quartiles, *p* = 0.012). In addition, increased baseline NETs levels in patients were associated with a 3.8-fold OD (95% CI:1.39–14.0, *p* = 0.009) in 30-day mortality [58]. These pathologies are accompanied by excessive NETosis, and NETs may be involved in damage to the alveolar endothelium and epithelium [59]. 

The pivotal role of neutrophils in the development of ARDS and the association of neutrophilia with lung injury in COVID-19 patients have already been described [60]. In these patients, neutrophil accumulation generates cytotoxic molecules that contribute to ARDS’s physiopathology, such as ROS, proteases, cytokines, and bioactive lipids, which lead to endothelial permeability and alveolar and lung injury [61]. A respiratory burst from activated neutrophils induces ROS release, such as superoxide radicals and H_2_O_2_, leading to oxidative stress, which contributes to the cytokine storm and blood clot formation in SARS-CoV-2 infection [46,62,63]. Therefore, excessive oxidative stress induced by neutrophil infiltration is related to alveolar damage, thrombosis, and severity in COVID-19 [63]. In this context, both the cytokine storm and ROS production, besides the infection by itself, can produce NETs formation (for more detail, see a previous review [64], in which we explain different NETs inducers and NETosis mechanisms, in addition to the role of neutrophils and NETs in thrombosis). Interestingly, SARS-CoV-2 can directly induce the release of NETs by healthy neutrophils in vitro, mechanistically depending on ACE2, serine protease, protein arginine deiminase 4 (PAD4), and virus replication [65].

Regarding ARDS, the DNA, histones, neutrophil elastase (NE), myeloperoxidase (MPO), and cathepsin G released during the NETosis process are cytotoxic to lung epithelial and endothelial cells [66]. DNA strands released during NETosis have been associated with diffuse alveolar damage and hemorrhage [67]. Histones present in the scaffold of NETs are harmful to endothelial cells, inducing an increase in endothelium permeability [68]. NE disrupts the cytoskeleton of endothelial cells, triggering its decomposition, and it induces E-cadherin and VE-cadherin impairment and apoptosis in epithelial cells, thus affecting the integrity of the alveolar barrier [66]. NE is associated with the processes of inflammation and thrombosis by way of the release of proinflammatory cytokines [69] and the proteolytic cleavage of the tissue factor pathway inhibitor (TFPI), thus enhancing the activation of factor X (FX) [70]. In addition, MPO leads to apoptosis of epithelial cells by means of ROS release [71]. In ARDS, like in other pathologies, other cells, such as platelets, also participate in NETs release by means of the binding of platelets and neutrophils through the toll-like receptor 4 (TLR4) and high-mobility group box 1 (HMGB1) expressed on their respective surfaces [52]. Consequently, neutrophils are activated, with the subsequent release of NETs, especially in pulmonary capillaries, resulting in vessel occlusion and severe lung injury (Figure 1) [52]. During immunothrombosis, DNA filaments, along with extracellular histones and neutrophil proteins, can create a scaffold in which platelets are trapped in the lung microcirculation. The increased endothelial permeability, together with microcirculation obstruction and immunothrombosis, promotes lung injury [72]. Moreover, it has been demonstrated in an in vitro model that the release of NETs by SARS-CoV-2-activated neutrophils promotes lung epithelial cell death [65]. In this context, one study which included 75 critical patients and seven healthy subjects demonstrated that, during the course of COVID-19, the levels of NETs remained high in samples from the lower respiratory tract. In keeping with this, NETs were detected in lung tissue from deceased COVID-19 patients, specifically in the bronchi and alveolar spaces [73]. The same study showed that the levels of NETs biomarkers, specifically of MPO-DNA complexes, correlated both with viral load in sputum samples (Spearman r correlation: *r* = 0.16, *p* = 0.009) and with neutrophil-recruiting chemokines (CXCL10) in blood (Spearman r correlation: *r* = 0.259, *p* = 0.026) [73]. Thus, it would seem that NETs are released and retained in the lower respiratory tract of critical COVID-19 patients, hence contributing to SARS-CoV-2-induced ARDS condition [73]. 

Accordingly, NETs have also been described as potential biomarkers of COVID-19 prognosis [49,65,74,75,76,77,78,79]. Several studies have analyzed different typical markers of NETs release in COVID-19 patients, such as cell-free DNA (cfDNA), citrullinated histone 3 (citH3), MPO, NE, and complexes of these NETs components with DNA (for example MPO-DNA complexes). The levels of these markers, mainly from plasma and serum samples, but also from tracheal aspirate and lung autopsy tissues, were compared to those of healthy donors or those of patients in different stages of the disease. All the studies demonstrated that NETs had been induced in COVID-19 patients, but the crucial observation was that the highest levels of NETs were seen in the critical group, demonstrating that NETs correlate with the severity of the illness [74,75,80,81,82,83]. In contrast, one study performed in children showed no differences in the production of cytokines and NETs [84] which was unsurprising since children infected by SARS-CoV-2 are less likely to present severe illness compared with adults. Moreover, the study in question did not include children with severe and critical disease, and only 25% of them presented moderate disease. 

Studies performed in adults have shown that cfDNA, citH3, NE, and MPO-DNA complexes are higher in hospitalized adult patients receiving mechanical ventilation versus non-mechanical ventilation, and even more crucially, that these markers are associated with short-term mortality [74,80]. Furthermore, specifically cfDNA, citH3, and MPO levels were decreased in day-three patients who survived, while they remained stable (abnormally high) in patients who died [80]. Additionally, MPO was negatively associated with the number of days of severe hypoxemia during a period of 7 days (in a study which included 58 ICU patients, with a multivariable negative binomial regression, after adjustment for age, gender, lung CT scan lesions, simplified acute physiological and Charlson score, and time interval since the beginning of symptoms and ICU admission) [83]. In particular, MPO-DNA complexes were correlated with Sequential Organ Failure Assessment (SOFA) score (Spearman *r* = 0.398, *p* = 0.036) and were significantly higher in COVID-19 patients admitted to intensive care units, whereas partial pressure of alveolar oxygen /fraction of inspired oxygen (PaO_2_/FiO_2_) correlated inversely (Spearman *r* = −0.403, *p* = 0.034) in a study of 28 hospitalized patients (age mean = 55.1 years, 60% male, 64.3% with diabetes, 46.5% with hypertension, 39.3% with chronic lung disease, 53.6% with ARDS) [74,81]. NE and histone-DNA complexes were found to be associated with intensive care admission, body temperature, lung damage, markers of cardiovascular outcomes, and renal failure. Moreover, a value of NE > 593 ng/mL was found to be an independent predictor of multi-organ injury (AUROC: 0.876, 95% CI: 0.758–0.95, percentage of correct classification: 85%) in a study of 121 patients (median age: 76 years, 48% male, 30% with diabetes, 60.8% with hypertension, 6.8% with chronic obstructive pulmonary disease, 29.7% with cardiovascular disease, 27% with dyslipidemia, and 14.9% with cancer) (Table 1) [82]. All these data indicate that the degree of neutrophil activation is related to COVID-19 severity. In this context, elevated levels of the anti-NETs IgG and IgM were detected in patients hospitalized with COVID-19 when compared to healthy controls, and most notably were associated with impaired oxygen efficiency and requirement of mechanical ventilation [85]. Importantly, these evaluations were performed in plasma or serum, clearly indicating that circulating NETs contribute to systemic inflammatory responses and are not likely to result from mechanical ventilation-induced pulmonary stress, a condition that is also associated with high levels of NETs [86].

Thrombosis is a critical complication in COVID-19 patients, and the role of NETs in this pathology has been described extensively [64]. In this regard, COVID-19-associated thrombosis presents higher plasma levels of NETs biomarkers, such as cfDNA, citH3, and MPO-DNA, and NETs structures have also been identified in arteriolar microthrombi [73,76,77,79,81,87,88]. In this context, MPO-DNA complexes (>0.12 OD, AUROC: 0.769, *p* < 0.001) and citH3 (>3.9 ng/mL, AUROC: 0.791, *p* < 0.001) provided moderate accuracy in identifying COVID-19 patients who would develop venous thromboembolism (VTE) in a study including 36 patients (age mean: 70.6 years, 55.5% male, 16.7% with diabetes, 41.67% with hypertension, 13.9% with atrial fibrillation, 5.56% with stroke, 5.6% with peripheral artery disease; 22.2% of whom developed VTE; 77.8% treated with low-molecular-weight heparin, 19.4% treated with antihypertensive agents, 13.9% treated with aspirin, 11.1% treated with statins, 8.3% treated with apixaban, and 5.6% treated with anti P2Y_12_) [81]. Furthermore, a positive correlation between cfDNA (Spearman *r* = 474, *p* < 0.001) and MPO-DNA complexes (Spearman *r* = 0.316, *p* = 0.002) with fibrinogen degradation products (a coagulation indicator) was observed in COVID-19 patients in a study including 19 mild and 41 severe patients (60 patients in total) [89]. Elevated levels of other coagulation indicators, such as tissue factor (TF) and sC5b-9 (a soluble biomarker of complement activation), have been detected in COVID-19 patients [79]. The co-localization of TF and FXIIa with NETs suggests that NETs accumulation leads to the activation of intrinsic and extrinsic coagulation pathways [79,90]. In this sense, histological studies of lung and other organs from COVID-19 patients have revealed the obstruction of numerous microvessels due to NETs-rich aggregates that have also been associated with endothelial damage or dysfunction [77].

## 4. Therapeutic Approaches to COVID-19 That Target Neutrophil Extracellular Traps

Given the important role of NETs in the progression of the disease caused by SARS-CoV-2, it is of vital importance to focus on the development of drugs that target NETs. A possible approach is to treat these patients with deoxyribonucleases (DNases) to compensate the impaired degradation of NETs seen in severe cases due to the lower levels and activity of DNase I in these patients [82,91]. In this context, another study showed that accumulation of NETs is partially due to impaired NETs clearance by DNase, as DNase substitution improved NETs dissolution and reduced FXII activation in vitro [90]. In line with this finding, recombinant human DNase has been shown to be beneficial in reducing sputum viscosity and improving lung function [92]. In this sense, there are reports about the role of dornase alfa, a highly purified solution of recombinant human DNase I, as a potential therapy for COVID-19. Two case studies have shown that the administration of aerosolized dornase alfa reduces the fraction of inspired oxygen requirements, induces partial resolution of bilateral opacities, and improves overall outcome [93,94]. Moreover, a study with DNase-I-coated melanin-like nanospheres demonstrated that these spheres prevent COVID-19 progression by diminishing sepsis-associated NETosis dysregulation, neutrophil count, and the levels of NETs biomarkers in plasma of SARS-CoV-2 sepsis patients [91]. Together, these findings point to dornase alfa as a potential drug for the treatment of severe COVID-19 patients. Naturally, clinical trials are required to corroborate the efficacy of this drug in this context; indeed, several clinical trials have recently been registered, not only for dornase alfa, but also for other drugs that target different aspects of NETs release.

As we show in Table 2, we have selected COVID-19 clinical trials with drugs that: (1) inhibit neutrophil recruitment or activation (colchicine), (2) prevent NETs release (fostamatinib), (3) specifically block NETs compounds (NE by alvelestat, gasdermin D by disulfiram, and HMGB1 by metformin), (4) degrade NETs (dornase alfa, also known as pulmozyme), and (5) reduce ROS or act as an antioxidant (vitamin D, cholecalciferol, and N-acetylcysteine) (Figure 2). Thus, there are almost 100 registered clinical trials that include a drug related to inhibition of NETs, though only 19 are currently complete or terminated (NCT04402970, NCT04527562, NCT04392141, NCT04667780, NCT04350320, NCT04326790, NCT04328480, NCT04867226, NCT04322682, NCT04625985, NCT04449718, NCT04400890, NCT04793243, NCT04483635, NCT04407286, NCT04733625, NCT04344041, NCT04419025, NCT04579393). However, only two of them, NTC04402970 and NTC04392141, have published results. NTC04402970, which analyzed dornase alfa efficacy, has suggested that the drug seems to improve mortality rates and reduce the length of ICU stays for COVID-19 patients; however, the data are preliminary and do not include any statistical analysis. NTC04392141, which tested the effects of colchicine, has provided data that suggest this drug improves mortality rates, length of hospitalization, lymphocyte count, and serum lactate dehydrogenase when compared to the standard treatment. 

Although these results are promising, we need to wait for the definite results of other clinical trials to corroborate if any of these drugs can be used efficiently for the treatment of COVID-19. The 73 selected ongoing clinical trials are shown in Table 2; they are being performed in several groups of patients, ranging from asymptomatic to mild (outpatients clinics), severe (hospitalized), and critical (ICU) patients. Over 50% of these trials are based on hospitalized patients, while under 10% of them include ICU patients, and only three trials include asymptomatic patients. On the other hand, some of the trials specify additional conditions related to cardiac injury, vitamin D, overweight, and other risk factors. The outcome measures of all the named clinical trials include mortality, disease severity, and disease progression, measured as length of hospital and/or ICU stay, change in oxygen pressure, SOFA score, and need for mechanical ventilation. Additionally, some of the trials include specific aspects including viral load and biomarkers—related to inflammation (CRP, interleukins), coagulation (D-dimer), or cardiovascular damage (troponin, CK) —, which are shown in Table 2. The trials emphasize the importance of studying both the reversal of severity and the recovery of critical patients admitted into hospital, and they highlight the relevance of the prevention of admissions of the outpatients or asymptomatic patients before the pulmonary infiltrate settles. Furthermore, while every one of the selected clinical trials includes adult patients, some of them also evaluate child patients (NCT04381936, NCT04402944, NCT04552951, NCT04502667, NCT04621058), and one of them includes pregnant women (NCT04825093). 

Regarding the clinical trials that are ongoing and shown in Table 2, most of them (52%) employ drugs to reduce ROS production and, thus, inhibit the pathway that triggers NETs release. In this group of trials, three different drugs are being employed as treatment; namely, cholecalciferol (vitamin D3 form), vitamin D, and N-acetylcysteine. Most (around 87%) use vitamin D or cholecalciferol. The effect of these drugs, independently of ROS reduction, could be related to the capacity of vitamin D to improve the innate and adaptative immune response, thus ameliorating the function of macrophages and dendritic cells [95,96,97,98]. Vitamin D promotes the production of cathelicidins and defensins, which attack the enveloped virus and undermine the production of pro-inflammatory mediators, consequently reducing the risk of the cytokine storm. Indeed, the deficiency of vitamin D (<50 nmol/L) is a potential risk factor for acute respiratory disorders, including viral infections [99,100,101]. In line with this, vitamin D deficiency has also been associated with COVID-19 disease severity [99,102]. N-acetylcysteine, a precursor of the antioxidant glutathione, has been used to loosen thick mucus in the lungs. However, N-acetylcysteine can also boost the immune system, suppress viral replication, and reduce inflammation [103]. Despite these benefits, N-acetylcysteine has been generally overlooked during the SARS-CoV and MERS-CoV epidemics, and throughout the COVID-19 pandemic [103]. The use of cholecalciferol, vitamin D, and N-acetylcysteine is being studied in ICUs, outpatients clinics, and even in asymptomatic and PCR-negative patients, and vitamin D deficiency has been added as an inclusion criteria in some of them. Viral load or viral clearance (time to negative PCR), and cardiac (troponin, CK), coagulatory (D-dimer), and/or inflammatory (CRP, interleukins) biomarkers are being analyzed. Others have recruited PCR-negative patients with the aim of analyzing the prevention of the infection and establishing the usefulness of vitamin D as prophylaxis treatment for the infection.

Another drug widely used in the clinical trials (almost 30%) we refer to is colchicine. Colchicine, widely used in auto-immune and inflammatory disorders, neutralizes the assembly of the NLR family pyrin domain containing 3 (NLRP3) inflammasome, reducing the release of IL-1β, IL-6 and other interleukins formed in response to danger signals [104,105]. Most of these clinical trials have recruited hospitalized patients, including moderate-to-severe ones, with specific conditions such as pneumonia, or those not requiring mechanical ventilation. Some of the colchicine clinical trials measure viral load or viral clearance, cardiac, coagulant and/or inflammatory biomarkers, and renal and thrombosis complications. Two of the colchicine trials have been suspended because the randomized evaluation of COVID-19 therapy provided no convincing evidence that further recruitment would provide conclusive proof of worthwhile benefit for the evaluation of colchicine in patients with COVID-19.

The rest of the clinical trials are using dornase alfa, metformin, fostamatinib, disulfiram, and alvelestat. Dornase alfa is currently employed as a mucolytic for the treatment of pulmonary disease. Metformin is one of the drugs most commonly used in diabetic patients. Fostamatinib is a spleen tyrosine kinase (SYK) inhibitor employed in the treatment of rheumatoid arthritis and immune thrombocytopenia purpura. Disulfiram is a drug employed for treating alcohol addiction, whose mechanism is based on the inhibition of pore formation by gasdermin D, allowing IL-1β and gasdermin D processing but prevents pore formation, thus preventing IL-1β release and pyroptosis. Alvelestat was developed for the treatment of lung diseases like chronic obstructive pulmonary disease, and works by blocking NE, which is responsible for inflammation and damage to the lungs. The role of these drugs in inhibiting neutrophil activation and NETs release provides new therapeutic indications for repurposing them to counteract inflammation and lung damage, both of which are important factors in COVID-19 severity.

## 5. Conclusions

SARS-CoV-2 infection can result in an exaggerated systemic inflammation, or cytokine storm, which triggers ARDS and multiple organ failure. This is one of the reasons why this disease is causing high rates of deaths and has become a global challenge for humanity. We have provided clinical data that illustrate the diversity of the biomarkers associated with COVID-19 disease progression, which can be hematologic, biochemical, coagulatory, or inflammatory. Furthermore, we have focused on neutrophils as the main cells leading to a prolonged inflammation status and infiltration of lung tissue. In this regard, neutrophilia, NLR, and NAR appear to predict COVID-19 severity, emphasizing the importance of neutrophils in this disease. NETs are also involved in ARDS, a common complication of COVID-19; therefore, NETs biomarkers, such as cfDNA, histones, MPO, and NE, among others, have also been described as potential forecasters of COVID-19 prognosis, specifically that associated with short-term mortality, ICU admission, mechanical ventilation, lung damage, and SOFA score.

Nevertheless, though predicting the disease’s prognosis is important, it is vital to find an effective treatment against COVID-19 if the pandemic is to be managed successfully. In this sense, we have analyzed the clinical trials registered to treat COVID-19 by employing drugs that target some step of NETs release and formation, for example, neutrophil activation (colchicine), NETs release (fostamatinib), DNA degradation (dornase alfa), NE (alvelestat), gasdermin D (disulfiram), HMGB1 (metformin), and ROS (cholecalciferol, N-acetylcysteine and vitamin D). Most of these trials are still ongoing at the time of writing, and should be completed in 2022. Therefore, while preclinical data indicate that the above are promising targets for COVID-19 treatment, we will have to see what the clinical data reveal in order to effectively address COVID-19 disease progression and chronification.

## Figures and Tables

**Figure 1 biomedicines-10-00031-f001:**
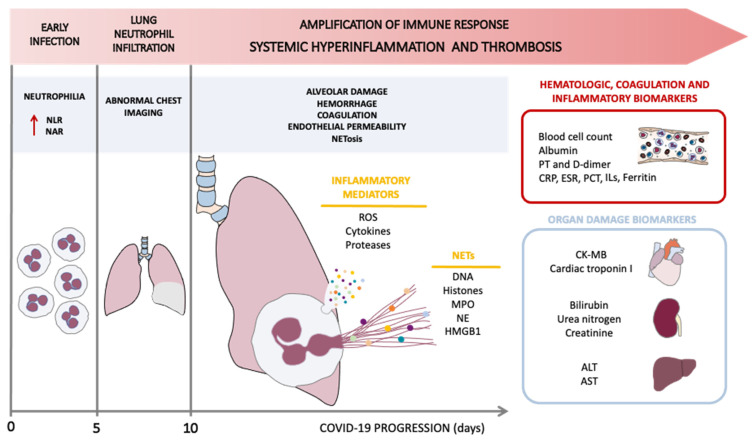
NETs involvement in the amplification of the systemic inflammatory and thrombotic response. SARS-CoV-2 infection presents different phases, accompanied by clinical biomarkers. Early infection, at 0–5 days, is characterized by neutrophilia and increases in the neutrophil-to-lymphocyte ratio (NRL) and neutrophil-to-albumin ratio (NAR). In approximately five days, patients present abnormal chest imaging (presenting ground-glass opacities of the lungs) with lung neutrophil infiltration. From approximately day 10, there is an amplification of immune-response-triggered systemic inflammation and thrombosis; this phase involves the release of inflammatory mediators, such as reactive oxygen species (ROS), cytokines, proteases, and neutrophil extracellular traps (NETs). This induces alveolar damage, hemorrhage, coagulation, endothelial permeability, and NETosis, increasing the severity of COVID-19. The following NETs components are released during NETosis: DNA, histones, myeloperoxidase (MPO), neutrophil elastase (NE), and high-mobility group box 1 (HMGB1) which is a protein that participates in platelet adhesion to NETs scaffolds and contributes to thrombus formation. PT: prothrombin time; CRP: C-reactive protein; ESR: erythrocyte sedimentation rate; PCT: procalcitonin; ILs: interleukins; CK-MB: creatine kinase-MB; ALT: alanine aminotransferase; AST: aspartate aminotransferase.

**Figure 2 biomedicines-10-00031-f002:**
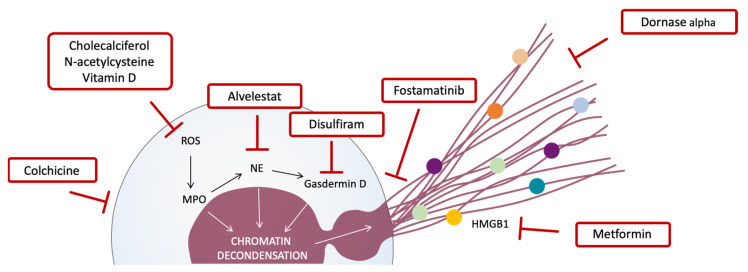
Potential drugs for COVID-19 treatment that target some step of NETs release and formation. Drugs targeting different steps of NETosis and NETs formation. Colchicine inhibits neutrophil activation. The presence of reactive oxygen species (ROS) is one of the first inducers of NETosis, and drugs such as cholecalciferol, N-acetylcysteine, and vitamin D can reduce ROS formation. ROS induce myeloperoxidase (MPO) activation, which activates neutrophil elastase (NE) and subsequently activates gasdermin D. MPO, NE, and gasdermin D promote chromatin decondensation, and gasdermin D also permeabilizes the membrane and creates pores in order to release DNA filaments and different proteins that compose NETs. In this regard, alvelestat and disulfiram block NE and gasdermin D, respectively; fostamatinib inhibits NETs release; dornase alfa degrades DNA filaments; and metformin inhibits HMGB1, a protein involved in platelet–NETs interactions, which triggers a loop of NETs release.

**Table 1 biomedicines-10-00031-t001:** Clinical signs associated with levels of different markers of neutrophil extracellular traps (NETs).

Clinical Signs	NETs Markers	References
Short-term mortality	cfDNA, citH3, NE and MPO-DNA complexes	[74,80]
Mechanical ventilation	cfDNA, citH3, NE and MPO-DNA complexes	[74,75,80,81]
Intensive care admission	NE, MPO- DNA and Histone-DNA complexes	[81,82]
Sequential Organ Failure Assessment score	MPO-DNA complexes	[81]
Lung damage	NE and Histone-DNA complexes	
Markers of cardiovascular outcomes		[82]
Renal failure		
Body temperature		
PaO_2_/fraction of inspired oxygen	MPO-DNA complexes *	[74]
Days with severe hypoxemia	MPO *	[83]

cfDNA: cell-free DNA; citH3: citrullinated histone 3; NE: neutrophil elastase; MPO: myeloperoxidase; PaO_2_: partial pressure of alveolar oxygen. * Inversely correlated.

**Table 2 biomedicines-10-00031-t002:** Ongoing clinical trials in COVID-19 patients with drugs that target a phase of the process of neutrophil extracellular trap (NETs) release.

Target	Drug	NTC Number	Patients					Outcome Measures		
			ICU	Hospitalized	Outpatients	Asymptomatic	Specific Conditions	VL/Clearance	Biomarkers	Others
**Neutrophil**	**Colchicine**	NCT04363437		X					Cardiac and Inflammatory	
		NCT04756128		X						
		NCT04492358		X						
		NCT04403243		X					Coagulatory and Inflammatory	
		NCT04367168		X						
		NCT04724629		X			Pneumonia		Coagulatory and Inflammatory	
		NCT04375202		X			Pneumonia			
		NCT04359095		X			Pneumonia			
		NCT04762771 ^†^		X			Cardiac injury		Cardiac and Inflammatory	
		NCT04510038 ^†^		X			Cardiac injury		Cardiac and Inflammatory	
		NCT04355143		X			Cardiac injury		Inflammatory	
		NCT04603690		X				X	Coagulatory and Inflammatory	
		NCT04360980		X				X	Inflammatory	
		NCT04818489		X					Inflammatory	
		NCT04472611		X						
		NCT04539873		X						
		NCT04381936 *		X						Renal and Thrombotic complications
		NCT04324463		X	X					
		NCT04416334			X		With risk factors			
		NCT04516941			X			X		
		NCT04322565			X	X	With risk factors			
**NETs release**	**Fostamatinib**	NCT04581954		X			Pneumonia			
		NCT04629703		X			Without respiratory failure and with risk factors			
**DNA**	**Dornase alfa**	NCT04355364	X	X	X		ARDS			
		NCT04359654		X			Systemic inflammation at risk of ventilatory failure		Inflammatory	
		NCT04432987	X	X						
		NCT04445285	X	X						
		NCT04488081	X	X						
		NCT04402944 *	X				Pneumonia			
**NE**	**Alvelestat**	NCT04539795			X				Inflammatory	
**Gasdermin D**	**Disulfiram**	NCT04594343		X					Coagulatory, Inflammatory and NETs	
		NCT04485130		X				X	Inflammatory	
**HMGB1**	**Metformin**	NCT04626089		X	X			X	Cardiac	
		NCT04604678			X					
		NCT04510194			X			X	Inflammatory	
**ROS**	**Cholecalciferol**	NCT04636086		X				X	Inflammatory	
		NCT04399746			X			X		
	**N-acetylcysteine**	NCT04755972	X							
		NCT04374461	X	X						
		NCT04792021		X					Inflammatory	
		NCT04703036		X					Inflammatory	
		NCT04928495			X			X	Cardiac, coagulatory and inflammatory	
	**Vitamin D**	NCT03188796	X				Severe Vitamin D defficiency			
		NCT04952857		X						
		NCT04395768		X						
		NCT04525820		X						
		NCT04552951 *		X				X	Cardiac, coagulatory and inflammatory	
		NCT04641195		X					Cardiac and inflammatory	
		NCT04502667 *		X					Coagulatory and inflammatory	
		NCT04621058 *					Vitamin D defficiency			
		NCT04385940		X	X		With risk factors		Inflammatory	
		NCT04334512			X					
		NCT04386850			X					
		NCT04780061			X					
		NCT04489628			X					
		NCT04868903			X					
		NCT04363840			X					
		NCT04482686			X				Inflammatory	
		NCT04334005			X					
		NCT04351490			X	X				
		NCT04536298			X					Infection prevention
		NCT04476680				X				
		NCT04482673			X		Negative/Positive PCR		Inflammatory	Infection prevention
		NCT04828538			X		Negative PCR/Symptomatic with risk factors			Infection prevention
		NCT04535791					Negative/Positive PCR HCW			Infection prevention
		NCT04372017					With high risk of contact and HCW			Infection prevention
		NCT04335084					With high risk of contact and HCW			Infection prevention
		NCT04596657					With high risk of contact and HCW			Infection prevention
		NCT04979065					Negative PCR HCW with overweight and obesity		Inflammatory	Infection prevention
		NCT04476745					Negative PCR with Vitamin D defficency		Inflammatory	
		NCT04825093					Pregnant			
		NCT04579640					Not specificed			
		NCT04734886					Not specificed		Inflammatory	

ICU: intensive care unit; VL: viral load; NETs: neutrophil extracellular traps; NE: neutrophil elastase; HMGB1: high-mobility group box 1; ROS: reactive oxygen species; HCW: healthcare workers. * Child patients. ^†^ Suspended.

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
