# Peer review of "Role of Neutrophil Extracellular Traps in COVID-19 Progression: An Insight for Effective Treatment"

_biomedicines, 2021, doi:10.3390/biomedicines10010031_

Round 1

Reviewer 1 Report

The review is well-written and covers an actual and very interesting topic related to COVID-19 pandemy and the role of NET formation by leukocytes. I only have a minor comment which is missing there:

  1. authors should explain the role of spike 2 (S2) subunit of the virus in the introduction (1-2 sentences). 

Author Response

Referee 1

“The review is well-written and covers an actual and very interesting topic related to COVID-19 pandemy and the role of NET formation by leukocytes”.

We very much appreciate that referee 1 considers that our manuscript covers an actual and very interesting topic related to COVID-19 pandemic and the role of NETs formation, that we provide a well-written manuscript considering both well organization and English use. The manuscript has been modified according to the referee’s suggestions. Thus, we have included the explanation of the role of spike 2 (S2) subunit of the virus.

Minor Comments:

 Point 1. “Authors should explain the role of spike 2 (S2) subunit of the virus in the introduction (1-2 sentences)”.

Response 1. According to referee’s suggestion, this explanation has been incorporated in the introduction (Pages 1, lines 38-40).

Reviewer 2 Report

This is a very interesting article, focusing on a topic of key current importance in COVID-19 therapy

My comments are directed to help making your paper have more impact on practitioners and researchers and more widely cited.

This is a review of the literature. Please provide your search terms and databases searched and languages searched so readers wishing to update your study later on can follow your methods. As this is not a systematic review a PRISMA statement is not required (unless you wish to provide one).

Introduction

“> 250 million cases and > 5 million deaths [please check the latest figures]

“the immune overreaction kills” [you have used this twice. No need to repeat]

Figure 1 HMGK1 high mobility group box 1 [please explain this in detail – some readers may not be familiar with it]

There are many statements of correlations but no strength of the correlations is presented. This is because you have not identified databases and analysed them in detail with numbers of patients, numbers of outcomes and statistical analyses. Can you find relevant databases in publications or by correspondence with authors? You will thereby be providing data of great importance to readers.

Examples are:

”abnormal chest imaging” [There is large literature – please provide specific diagnostic patterns]

“3.65 x 109 neutrophils/L”  “cut off value of NCR > 13 may help clinicians” [these are the rare occasions on which you present numerical data]

“lower lymphocyte correlation”

“ROC analysis with the sensitivity and specificity of 71.7% has recently related a NAR value of 207.5 with mortality in all patients with COVISD-19” [what is the sensitivity and what is the specificity?] [Please provide the other results and the other key values which relate to outcomes]  

“a recent clinical trial highlighted the correlation between levels of NETs and the severity of pneumonia” [this is crucial to your argument. Please present the key details of the study – patient numbers, sex, ages, comorbidities, therapies, NETs levels and how severity of pneumonia was measured. Outcomes? This sentence is typical of the brief statements unsupported by presentation of data that limit the value of your manuscript]

“neutrophil accumulation generates toxic molecules that might contribute to ARDs pathophysiology.” [might? – please explain. If your read the introduction of many articles which attempt to link pathophysiological constructs to clinical outcomes the words “might” or “may” occur frequently]

“in particular, MPO-DNA complexes were correlated with Sequential Organ Failure” [correlated? Strength of association? Sample size? Patients? comorbidities? therapies? …This is crucial to your argument. Please describe the findings on the Sequential Organ Failure Scale]

[There are other statements of correlation that the reader for which the reader would like to receive specific data to aid clinical care. I searched and noted 6 uses of the word may, 2 of might, 5 of could, 4 of correlated, and 6 of cause/causes/caused. Please support your assertions with data].

 Results

Section 4. Therapeutic approaches

This is the crucial contribution you are making.

“In fact, a transcriptomic profile study has demonstrated that neutrophil activation and degranulation are highly activated processes in SARS infection”

“Likewise, neutrophilia has been described as an indicator of severe respiratory symptoms and poor outcomes in patients with COVID-19.”

[More examples of imprecise statements. Please go over this section in detail and examine each statement and provide details of study design and quantitative outcomes.”]

The English text and syntax are excellent. Compliments to you and your English-speaking editor.

Typos: Abstract    result   change to    resulted

Billirrubin change to bilirubin

Author Response

Referee 2

This is a very interesting article, focusing on a topic of key current importance in COVID-19 therapy. My comments are directed to help making your paper have more impact on practitioners and researchers and more widely cited.”

We fully appreciate the comments of reviewer 2 and thank her/him for considering that “this is a very interesting article, focusing on a topic of key current importance in COVID-19 therapy”.  Care has been taken to modify the new version in accordance with his/her suggestions, which we believe have helped to improve our work, and will make our review has more impact and be more cited. According to his/her suggestion, a paragraph has been included in the introduction section to provide our search terms and databases, and details of study design, number of patients, comorbidities, outcomes, statistical analyses have been provided to the correlations statements in section 3 Neutrophils and Neutrophil extracellular traps in COVID-19. The specific responses to the comments of the referee are as follows:

 Point 1. This is a review of the literature. Please provide your search terms and databases searched and languages searched so readers wishing to update your study later on can follow your methods. As this is not a systematic review a PRISMA statement is not required (unless you wish to provide one).

Response 1. The databases, dates, language, and terms employed for the search have been provided in the last paragraph of the introduction (Page 2, lines 63-82).

Point 2. Introduction: “> 250 million cases and > 5 million deaths [please check the latest figures].

Response 2. The number of cases and deaths of COVID-19 has been updated in the abstract (Page 1, line 11) and in the introduction (Page 2, lines 54-55).

 Point 3. “the immune overreaction kills” [you have used this twice. No need to repeat]

Response 3. “the immune overreaction kills” has been removed in section 2. Biomarkers in COVID-19 (Page 2, line 87).

 Point 4. Figure 1 HMGB1 high mobility group box 1 [please explain this in detail – some readers may not be familiar with it]

Response 4. A description of HMGB1 has been inserted in the legend of Figure 1 (Page 3, line 130).

There are many statements of correlations but no strength of the correlations is presented. This is because you have not identified databases and analysed them in detail with numbers of patients, numbers of outcomes and statistical analyses. Can you find relevant databases in publications or by correspondence with authors? You will thereby be providing data of great importance to readers.

 We thank the reviewer by his/her suggestion to further develop the detailed data regarding numbers of patients, numbers of outcomes and statistical analyses. We have incorporated them in each particular case.

 Examples are:

 Point 5. ”abnormal chest imaging” [There is large literature – please provide specific diagnostic patterns]

Response 5. The specific diagnostic patterns for chest computed tomography have been provided in section 2. Biomarkers in COVID-19 and reference 22 has been included (Page 2, lines 96-100).

Point 6. “lower lymphocyte correlation”

Response 6. Details of numbers and characteristics of patients, cut off values and statistical analyses have been added to this statement in section 3. Neutrophils and Neutrophil extracellular traps in COVID-19 (Page 4, lines 163-175).

Point 7. “ROC analysis with the sensitivity and specificity of 71.7% has recently related a NAR value of 207.5 with mortality in all patients with COVID-19” [what is the sensitivity and what is the specificity?] [Please provide the other results and the other key values which relate to outcomes]

 Response 7. We have provided the number and some characteristics of patients included in this study, statistical data. Furthermore, we have specified the sensitivity and the specificity of the analysis (Page 5, lines 186-191).

Point 8. “a recent clinical trial highlighted the correlation between levels of NETs and the severity of pneumonia” [this is crucial to your argument. Please present the key details of the study – patient numbers, sex, ages, comorbidities, therapies, NETs levels and how severity of pneumonia was measured. Outcomes? This sentence is typical of the brief statements unsupported by presentation of data that limit the value of your manuscript]

Response 8.  We have included precise details of this study including patient numbers, ages, sex, comorbidities, therapies, details of how severity of pneumonia was measured, outcomes and statistical data and reference 59 has been incorporated (Pages 5-6, lines 222-233).

Point 9. “neutrophil accumulation generates toxic molecules that might contribute to ARDs pathophysiology.” [might? – please explain. If your read the introduction of many articles which attempt to link pathophysiological constructs to clinical outcomes the words “might” or “may” occur frequently]

Response 9. We have explained this sentence by providing some mediators such as ROS, proteases, cytokines and bioactive lipids, which are released by extravasated neutrophils and are involved in endothelial permeability and alveolar and lung injury (Page 6, lines 238-240).

Point 10. “in particular, MPO-DNA complexes were correlated with Sequential Organ Failure” [correlated? Strength of association? Sample size? Patients? comorbidities? therapies? …This is crucial to your argument. Please describe the findings on the Sequential Organ Failure Scale]

Response 10. We have added to this statement information regarding the number of patients included in this analysis, mean age, sex, comorbidities, Spearman r and statistical data for both SOFA and PaO2/FiO2 correlation (Page 7, lines 314-319).

Point 11. [There are other statements of correlation that the reader for which the reader would like to receive specific data to aid clinical care. I searched and noted 6 uses of the word may, 2 of might, 5 of could, 4 of correlated, and 6 of cause/causes/caused. Please support your assertions with data].

Response 11. Following referee’s suggestion, we have provided data to support our statement correlations. We have added number of patients, age, sex, comorbidities and therapies, and statistical analysis and/or statistical data to the following correlation statements of the manuscript:

  • “The same study also showed that viral load in sputum samples and neutrophil-recruiting chemokines in blood correlate with plasma NETs biomarkers levels in plasma samples” (Page 6, lines 276-280).
  • “MPO was negatively associated with the number of days of severe hypoxemia” (Page 7, lines 303-312)
  • “NE and histone-DNA complexes were associated with intensive care admission, body temperature, lung damage, markers of cardiovascular outcomes and renal failure, besides NE > 593 ng/mL was an independent predictor of multi-organ injury” (Page 7, lines 314-319).
  • “MPO-DNA complexes (>0.12 OD) and citH3 (>3.9 ng/mL) provide moderate accuracy in identifying COVID-19 patients who will develop venous thromboembolism (VTE)” (Page 7-8, lines 332-339).
  • “a positive correlation between cfDNA and MPO-DNA complexes with fibrinogen degradation products (a coagulation indicator) has been observed in COVID-19 patients” (Page 8, lines 340-343).

Results

Section 4. Therapeutic approaches

This is the crucial contribution you are making.

Point 12. “In fact, a transcriptomic profile study has demonstrated that neutrophil activation and degranulation are highly activated processes in SARS infection”

Response 12. We have provided supporting data for this transcriptomic study including the dataset on which the analysis is based, and the GO terms obtained that justify the assertion (Page 4, lines 149-152).

Point 13. “Likewise, neutrophilia has been described as an indicator of severe respiratory symptoms and poor outcomes in patients with COVID-19.”

Response 13. The manuscript has been modified to include the cut off values of neutrophil counts that can predict COVID-19 severity, number and characteristics of patients, statistical analyses, statistical data and the values of sensitivity and specificity (Page 4, lines 154-161).

Point 14. [More examples of imprecise statements. Please go over this section in detail and examine each statement and provide details of study design and quantitative outcomes.”]

Response 14. We have gone over the section in detail and we have specified the rest of the statements that we have supported with detailed data regarding number of patients, age, sex, comorbidities, therapies, statistical analysis and/or statistical data in the response to the point 11.

The English text and syntax are excellent. Compliments to you and your English-speaking editor.

We thank very much referee 2 for considering that the English text and syntax are excellent.

Point 15. Typos: Abstract    result   change to    resulted

Billirrubin change to bilirubin

Response 15. The typos have been corrected in abstract (Page 1, line 10) and Figure 1 (Page 3, lines 120).

Round 2

Reviewer 2 Report

Reading the authors' responses they have replied to all my concerns.